# Unraveling the Complexity of Memory in RL Agents: an Approach for Classification and Evaluation

**Egor Cherepanov**[1,2]**, Nikita Kachaev**[1,3]**, Artem Zholus**[4,5]**, Alexey K. Kovalev**[1,2]**, Aleksandr I. Panov**[1,2]
[1]AXXX, [2]MIRIAI, [3]ITMO University AI Talent Hub, [4]Mila – Quebec AI Institute, [5]Polytechnique Montréal
`cherepanov@axxx.tech`

## Abstract

The incorporation of memory into agents is essential for numerous tasks within the domain of Reinforcement Learning (RL). In particular, memory is paramount for tasks that require the use of past information, adaptation to novel environments, and improved sample efficiency. However, the term "memory" encompasses a wide range of concepts, which, coupled with the lack of a unified methodology for validating an agent's memory, leads to erroneous judgments about agents' memory capabilities and prevents objective comparison with other memory-enhanced agents. This paper aims to streamline the concept of memory in RL by providing practical precise definitions of agent memory types, such as long-term vs. short-term memory and declarative vs. procedural memory, inspired by cognitive science. Using these definitions, we categorize different classes of agent memory, propose a robust experimental methodology for evaluating the memory capabilities of RL agents, and standardize evaluations. Furthermore, we empirically demonstrate the importance of adhering to the proposed methodology when evaluating different types of agent memory by conducting experiments with different RL agents and what its violation leads to.

## 1 Introduction

Reinforcement Learning (RL) effectively addresses problems within the Markov Decision Process (MDP) framework. However, applying RL to tasks with partial observability remains challenging, requiring agents to efficiently process their interaction history (Esslinger et al., 2022; Hausknecht & Stone, 2015; Ni et al., 2021).

In complex environments with noisy observations and long episodes, storing and retrieving important information becomes crucial (Goyal et al., 2022; Graves et al., 2016). Yet, the concept of *"memory"* in RL literature lacks unified definition. Some works define it as the ability to **handle dependencies within a fixed context** (Esslinger et al., 2022; Ni et al., 2023), others as the ability to **use out-of-context information** (Parisotto et al., 2020), and in Meta-RL, as the ability to **adapt to new environments** (Team et al., 2023).

However, in the absence of clear definitions and standardized evaluation protocols, claims about memory capacity in RL agents remain vague and often misleading. Memory is frequently attributed to architectural features like recurrence or attention, yet without proper isolation of memory effects, such assumptions can be incorrect. For instance, an agent might appear to exhibit long-term memory simply due to task configurations that allow shortcuts or overlap with short-term context. As a result, many empirical evaluations risk conflating different memory mechanisms or failing to detect architectural limitations. This hinders progress in developing truly memory-capable agents and comparing models in a fair and reproducible manner.

In this work, we attempt to unify and clarify the concept of memory in RL agents by treating memory as an intrinsic attribute of memory-enhanced agents, directly linking memory type classification to the agent's internal mechanisms. These specific memory types - short-term vs. long-term and declarative vs. procedural - can be rigorously assessed through experiments in memory-intensive environments. Our classification, based on temporal dependencies and the nature of the recalled infor-

mation, provides a structured framework for distinguishing memory types, enabling fair comparisons, diagnosing architectural limitations, and guiding principled improvements.

It is important to clarify that our goal is not to replicate the full complexity of human memory. Rather, we selectively adapt well-established memory concepts from neuroscience - such as short-term, long-term, declarative, and procedural memory - that are already informally used in RL, but lack precise definitions and formal grounding (Fortunato et al., 2020; Kang et al., 2024b; Ni et al., 2023).

In summary, our main contributions are as follows:

1. We provide formal definitions of key memory types in RL – specifically, *short-term (STM) vs. long-term (LTM)* and *declarative vs. procedural* memory – grounded in neuroscience and formalized for RL settings – Section 4.

2. We introduce a task-level decoupling of *Memory Decision-Making (Memory DM)* and *Meta-RL*, clarifying the behavioral role of memory in each category – Section 4.

3. We propose a principled experimental methodology for evaluating STM and LTM in Memory DM tasks, including precise criteria for identifying memory boundaries – Section 4.2.

4. We show that neglecting the proposed methodology can mislead conclusions about agent memory capabilities, highlighting the importance of proper experimental configuration – Section 5.

## 2 BACKGROUND

### 2.1 MEMORY OF HUMANS AND AGENTS

Many RL studies reference memory types from cognitive science, such as long-term (Lampinen et al., 2021; Ni et al., 2023), working (Graves et al., 2014), associative (Polson, 1975), and episodic memory (Pritzel et al., 2017), but often reduce them to vague temporal categories (e.g., short vs. long-term), with short-term spanning a few steps and long-term hundreds. This oversimplification, ignoring the relative nature of memory, complicates meaningful evaluation. To resolve this, we formalize agent memory types and introduce a principled evaluation framework.

#### 2.1.1 MEMORY IN COGNITIVE SCIENCE

Human adaptive behavior depends heavily on memory, which governs how knowledge and skills are acquired, retained, and reused (Parr et al., 2020; 2022). Memory exists in many forms, each of which relies on different neural mechanisms. Neuroscience and cognitive psychology distinguish memory by the temporal scales at which information is stored and accessed, and by the type of information that is stored. Abstracting from this distinction, a high-level definition of human memory is as follows: **"*memory – is the ability to retain information and recall it at a later time*"**.

This definition aligns with how memory is typically understood in RL, and we adopt it to define types of RL agent memory. In neuroscience, memory is classified by timescale and behavioral function, distinguishing *short-term* memory, lasting seconds, from *long-term* memory, which can persist for a lifetime (Davis & Squire, 1984). It is also divided into *declarative* (explicit) and *procedural* (implicit) forms (Graf & Schacter, 1985): the former involves consciously recalled facts and events, while the latter relates to unconscious skills like riding a bike or skiing. Though these distinctions are well-established in neuroscience, RL requires precise, testable definitions. In what follows, we adapt these cognitive categories into a formal framework suitable for RL agents.

#### 2.1.2 MEMORY IN RL

Memory in RL encompasses diverse agent capabilities, but its definition varies across studies. In many POMDPs, agents must retain key information to act effectively later within the same environment. This typically involves two kinds of temporal dependencies: 1) within a bounded time window (e.g., transformer context (Esslinger et al., 2022; Ni et al., 2023; Yue et al., 2024)); 2) beyond the current context, requiring persistent storage or recall (Parisotto et al., 2020; Sorokin et al., 2022). As noted in Section 2.1.1, STM and LTM correspond to different temporal scopes of declarative memory (see Figure 1). In contrast, Meta-RL

involves procedural memory, enabling agents to reuse skills across tasks (Team et al., 2023) (see Figure 2). However, many works conflate these types, evaluating "long-term memory" solely in Meta-RL settings based on MDPs (Kang et al., 2024a), without isolating decision-making from past information. To resolve this, we formalize RL memory types by task structure and temporal dependencies. In this work, we focus on **declarative memory**, guiding decisions from past observations in the same environment, emphasizing its short- and long-term forms.

### 2.1.3 MEMORY AND CREDIT ASSIGNMENT

Papers on agent memory, especially declarative memory, often distinguish between two forms of temporal reasoning: *memory* and *credit assignment* (Mesnard et al., 2020; Ni et al., 2023; Osband et al., 2019). In Ni et al. (2023), *memory* is defined as recalling a past event at the current time, while *credit assignment* is identifying when reward-relevant actions occurred. Though distinct, both concepts describe temporal dependencies between events. Here, we focus on the agent's ability to form such dependencies, treating memory and credit assignment as one. We adopt the general definition from Section 2.1.1, which applies to both, as it captures their shared temporal nature.

**Figure 1** (right column top):

**Long-term memory** $\xi > K$

event — agent's context

$t_e \quad t_e + \Delta t \qquad t_r - K + 1 \qquad \xi \qquad t_r$

**Short-term memory** $\xi \leq K$

event agent's context

$t_r - K + 1 \qquad t_e \quad t_e + \Delta t \qquad \xi \qquad t_r$

Figure 1: STM vs. LTM. $t_e$ - event start, $t_r$ - recall time; $K$ - context length, $\xi$ – correlation horizon. If the event lies beyond $K$, LTM is needed; if within, STM is enough.

## 3 RELATED WORKS

Interest in memory-enhanced RL has led to numerous architectures (Oh et al., 2016; Lampinen et al., 2021; Fortunato et al., 2020) and benchmarks (Morad et al., 2023a; Cherepanov et al., 2025; Osband et al., 2019; Pleines et al., 2023), yet the term "memory" remains inconsistently defined and often misaligned with what experiments actually test. Some works define memory as retaining recent observations within the same episode – either via recurrent states (Hausknecht & Stone, 2015), transformer contexts (Esslinger et al., 2022; Grigsby et al., 2024), or external stores (Lampinen et al., 2021; Le et al., 2024). Others expand it to long-range dependencies through learned state compression (Morad et al., 2023b), key-value recurrent updates (Pramanik et al., 2023; Cherepanov et al., 2024), or spatial memory maps (Parisotto & Salakhutdinov, 2017b). A separate line views memory as cross-episode knowledge transfer, e.g., in Meta-RL (Kang et al., 2024a; Bauer et al., 2023). This variety – from within-episode recall to multi-task adaptation – reflects the lack of a shared definition. Our work addresses this gap by introducing a unified taxonomy grounded in temporal dependencies and task structure.

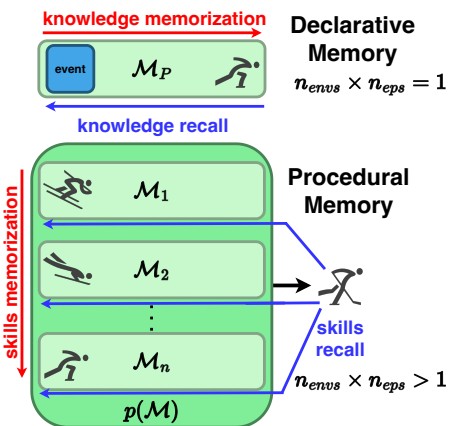

Figure 2: Illustration of declarative and procedural memory. Red arrows represent memorization steps, blue arrows indicate the recall of task-relevant information.

Among concrete instantiations, Esslinger et al. (2022) introduced Deep Transformer Q-Networks (DTQN), which leverage transformer context for partially observable RL. Ni et al. (2023) extended this line with GPT-2 based agents, including DQN-GPT-2 and SAC-GPT-2, which apply attention-based memory to online RL control. In the offline setting, Decision Transformer (DT) (Chen et al., 2021) uses a sequence model trained on trajectories for return-conditioned planning, while recurrent baselines such as BC-LSTM provide a contrast between attention-based and recurrent memory mechanisms. These models constitute the baselines we evaluate in our experiments.

Ni et al. (2023) further distinguish between memory – the ability to recall past events – and credit assignment – identifying when reward-relevant actions occurred. Kang et al. (2024b) build on reconstructive memory (Bartlett & Kintsch, 1995), emphasizing reflection grounded in interaction.

These varied interpretations underscore the need for a unified definition of memory in RL. We address this by formalizing memory types via temporal dependencies and task structure, and proposing a framework for empirical evaluation. Concurrently with our study, Yue et al. (2024) introduced memory dependency pairs $(p, q)$ to model recall in demonstrations. While insightful for imitation learning, their approach lacks a theoretical treatment of RL memory and does not address the broader taxonomy of agent memory types.

## 4 MEMORY IN RL

POMDP tasks involving memory fall into two categories: *Meta-RL*, focused on skill transfer across tasks, and *Memory DM*, where agents recall past information for future decisions. This distinction matters: Meta-RL relies on procedural memory for rapid adaptation, while Memory DM uses declarative memory to guide decisions within a single environment. Yet many works reduce memory to temporal range, ignoring the behavioral roles that distinguish these types. To formalize Memory DM tasks, we first define the agent's context length:

**Definition 4.1.** *Agent context length ($K \in \mathbb{N}$) – is the maximum number of previous steps (triplets of $(o, a, r)$) that the agent can process at time $t$.*

For example, an MLP-based agent processes one step at a time ($K = 1$), while a transformer-based agent can process a sequence of up to $K = K_{attn}$ triplets, where $K_{attn}$ is determined by attention. Looking ahead, RNNs also have a $K = 1$, but using hidden states allows longer dependencies to be handled. Using the introduced Definition 4.1 for agent context length, we can introduce a formal definition for the Memory DM framework we focus on in this paper:

**Definition 4.2.** *Memory Decision-Making (Memory DM) – is a class of POMDPs in which the agents decision-making process at time $t$ is based on the history $h_{0:t-1} = \{(o_i, a_i, r_i)\}_{i=0}^{t-1}$ if $t > 0$ otherwise $h = \varnothing$. The objective is to determine an optimal policy $\pi^*(a_t \mid o_t, h_{0:t-1})$ that maps the current observation $o_t$ and history $h_{0:t-1}$ of length $t$ to an action $a_t$, maximizing the expected cumulative reward within a single POMDP*

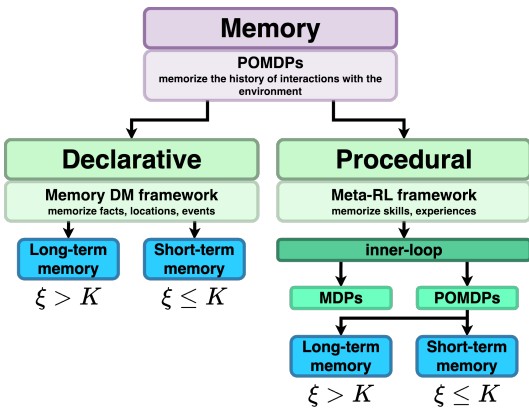

Figure 3: Classification of memory types of RL agents. While the Memory DM framework contrasts with Meta-RL, its formalism can also describe inner-loop tasks when they are POMDPs.

*environment $\mathcal{M}_P$: $J^\pi = \mathbb{E}_\pi \left[ \sum_{t=0}^{T-1} \gamma^t r_t \right]$, where $T$ – episode duration, $\gamma \in [0, 1]$ – discount factor.*

In the Memory DM framework (Definition 4.2), memory refers to the agent's ability to recall information from the past within a single environment and episode. In contrast, in the Meta-RL framework (Definition 4.3), memory involves recalling information about the agent's behavior from other environments or previous episodes:

**Definition 4.3.** *Meta-RL – is a class of POMDPs where the agent learns to learn from its past experiences across multiple tasks and memorize the common patterns and structures to facilitate efficient adaptation to new tasks. Let $\mathcal{D} = \{\tau_j^{\mathcal{M}_i}\}_{j=0}^{H-1}$ is all of the data of $H$ episodes of length $T$ collected in the MDP $\mathcal{M}_i \sim p(\mathcal{M})$. A Meta-RL algorithm is a function $f_\theta$ that maps the data $\mathcal{D}$ to a policy $\pi_\phi$, where $\phi = f_\theta(\mathcal{D})$. The objective to determine an optimal $f_\theta$: $J^\theta = \mathbb{E}_{\mathcal{M}_i \sim p(\mathcal{M})} \left[ \mathbb{E}_{\mathcal{D}} \left[ \sum_{\tau \in \mathcal{D}_{I:H}} G_i(\tau) \Big| f_\theta, \mathcal{M}_i \right] \right]$, where $G_i(\tau)$ – discounted return in the MDP $\mathcal{M}_i$, $I$ – index of the first episode during the trial in which return counts towards the objective (Beck et al., 2024).*

To operationalize the distinction between memory types in RL, we translate the neuroscience concepts of declarative and procedural memory (Section 2.1.1) into measurable task-level criteria:

**Definition 4.4 (Declarative and Procedural memory in RL).** *Let $n_{envs}$ be the number of training environments and $n_{eps}$ the number of episodes per environment. Then,*

1. **Declarative Memory** – *a type of agent memory when an agent transfers its knowledge within a single environment and across a single episode within that environment:*

$$\text{Declarative Memory} \iff n_{envs} \times n_{eps} = 1. \tag{1}$$

2. **Procedural Memory** – *a type of agent memory when an agent transfers its skills across multiple environments or multiple episodes within a single environment:*

$$\text{Procedural Memory} \iff n_{envs} \times n_{eps} > 1. \tag{2}$$

In this formulation, *knowledge* refers to observable, environment-specific information – such as object locations or facts – used within a single episode. *Skills*, in contrast, are policies reused across tasks or trials. Accordingly, Memory DM primarily evaluates declarative memory, while Meta-RL settings test procedural memory (see Figure 3).

Having distinguished declarative and procedural memory, we now examine the temporal structure of memory in the Memory DM framework, focusing on its division into short-term and long-term forms.

**Definition 4.5 (Memory DM types of memory).** *Let $K$ be the agent context length, $\alpha_{t_e}^{\Delta t} = \{o_i, a_i, r_i\}_{i=t_e}^{t_e+\Delta t}$ – an event of duration $\Delta t$ that begins at $t = t_e$ and ends at $t = t_e + \Delta t$, and $\beta_{t_r}(\alpha_{t_e}^{\Delta t}) = a_t \mid (o_t, \alpha_{t_e}^{\Delta t})$ – a decision-making point (recall) at time $t = t_r$ based on the current observation $o_t$ and information about the event $\alpha_{t_e}^{\Delta t}$. Let also $\xi = t_r - t_e - \Delta t + 1$ be the* **correlation horizon***, i.e. the minimal time delay between the event $\alpha_{t_e}^{\Delta t}$ that supports the decision-making and the moment of recall of this event $\beta_{t_r}$. Then,*

1. **Short-term memory (STM)** - *an agent's ability to use information about local correlations from the past within the context of length $K$ at decision time:*

$$\beta_{t_r}(\alpha_{t_e}^{\Delta t}) = a_t \mid (o_t, \alpha_{t_e}^{\Delta t}) \, \forall \, \xi = t_r - t_e - \Delta t + 1 \leq K.$$

2. **Long-term memory (LTM)** - *an agent ability to utilize information about global correlations from the past outside of the agent context of length $K$, during decision-making:*

$$\beta_{t_r}(\alpha_{t_e}^{\Delta t}) = a_t \mid (o_t, \alpha_{t_e}^{\Delta t}) \, \forall \, \xi = t_r - t_e - \Delta t + 1 > K.$$

*An illustration for the definitions of classifying Memory DM tasks into LTM and STM from Definition 4.5 is shown in Figure 1.*

The two definitions of declarative memory encompass all work related to Memory DM tasks, where decisions are based on past information. Meta-RL consists of an inner-loop, where the agent interacts with the environment $\mathcal{M} \sim p(\mathcal{M})$, and an outer-loop for transferring knowledge between tasks. Typically, $\mathcal{M}$ is an MDP that doesn't require memory, serving only the outer-loop, which is what "memory" refers to in Meta-RL studies.

The tasks in which the agent makes decisions based on interaction histories in the inner-loop are not named separately, since the classification of Meta-RL task types (multi-task, multi-task zero-shot, and single-task) is based solely on outer-loop parameters ($n_{envs}$ and $n_{eps}$) and does not consider inner-loop task types. However, we can classify the agent's memory for these tasks as declarative STM or LTM (Figure 3).

We introduce an additional decoupling of Meta-RL task types into green (with POMDP inner-loop tasks) and blue (with MDP inner-loop tasks). In the green case, the agent's memory is required for both skill transfer in the outer-loop and decision-making from interaction histories in the inner-loop, and within the inner-loop can be considered a Memory DM. In the blue case, memory is needed only for skill transfer. While this paper focuses on Memory DM tasks, the terminology enables further classification of Meta-RL tasks, with POMDP sub-classes highlighted in green. The proposed classification of tasks requiring agent memory is shown in Table 1.

## 4.1 MEMORY-INTENSIVE ENVIRONMENTS

To effectively test a Memory DM agent's use of short-term and long-term memory, it is crucial to design appropriate experiments. Not all environments are suitable for assessing agent memory; for

example, omnipresent Atari games (Bellemare et al., 2013) with frame stacking or MuJoCo control tasks (Fu et al., 2021) may yield unrepresentative results. To facilitate the evaluation of agent memory capabilities, we formalize the definition of memory-intensive environments:

**Definition 4.6 (Memory-Intensive Environments).** *Let $\mathcal{M}_P$ be a POMDP, and let $\Xi = \{\xi_n\}_n = \{(t_r - t_e - \Delta t + 1)_n\}_n$ denote the set of correlation horizons for all event-recall pairs $(\alpha_{t_e}^{\Delta t}, \beta_{t_r})$. Then $\mathcal{M}_P$ is a memory-intensive environment, denoted $\tilde{\mathcal{M}}_P$, if and only if: $\min_n \xi_n > 1$.*

**Corollary 1.** *A task corresponds to an MDP (i.e., is Markovian) if and only if all correlation horizons are trivial: $\max_n \Xi = 1$.*

*Proof.* In an MDP, the optimal action depends only on the current state (or observation), i.e., no past information

Table 1: Classification of tasks requiring agent memory based on our definitions: green indicates tasks described by the proposed definitions of LTM and STM, while blue indicates those that are not. Meta-RL tasks with a POMDP inner-loop are marked green as they can be classified as Memory DM tasks. POMDP† indicates a Memory DM task considered as an inner-loop task without an outer-loop.

| $n_{envs}$ | $n_{eps}$ | POMDP | Inner-loop task | Memory | Tasks that require agent memory | |
|---|---|---|---|---|---|---|
| | | | | | **Memory DM** | |
| | | | | | LTM $\xi > K$ | STM $\xi \le K$ |
| 1 | 1 | Memory DM | POMDP† | Dec. | Long-term memory task | Short-term memory task |
| | | | | | **Meta-RL: Outer-loop and inner-loop memory** | |
| | | | | | LTM $\xi > K$ | STM $\xi \le K$ |
| 1 | >1 | Meta-RL | POMDP | Proc. | Single-task | Single-task |
| >1 | 1 | Meta-RL | POMDP | Proc. | Multi-task zero-shot | Multi-task zero-shot |
| >1 | >1 | Meta-RL | POMDP | Proc. | Multi-task | Multi-task |
| | | | | | **Meta-RL: Outer-loop memory only** | |
| | | | | | No memory $\xi = 1$ | No memory $\xi = 1$ |
| 1 | >1 | Meta-RL | MDP | Proc. | Single-task | Single-task |
| >1 | 1 | Meta-RL | MDP | Proc. | Multi-task zero-shot | Multi-task zero-shot |
| >1 | >1 | Meta-RL | MDP | Proc. | Multi-task | Multi-task |

is needed. This implies $\xi_n = 1$ for all event-recall pairs, hence $\max_n \xi_n = 1$. Conversely, if $\max_n \xi_n = 1$, then no decision depends on events beyond the current step, satisfying the Markov property. ∎

Using the definitions of memory-intensive environments (Definition 4.6) and agent memory types (Definition 4.5), we can configure experiments to test short-term and long-term memory in the Memory DM framework. Notably, the same memory-intensive environment can validate both types of memory, as outlined in Theorem 2:

**Theorem 2 (On the context memory border).** *Let $\tilde{\mathcal{M}}_P$ be a memory-intensive environment and $K$ be an agent's context length. Then there exists context memory border $\overline{K} \ge 1$ such that if $K \le \overline{K}$ then the environment $\tilde{\mathcal{M}}_P$ is used to validate exclusively long-term memory in Memory DM framework:*

$$\exists \, \overline{K} \ge 1 : \forall \, K \in [1, \overline{K}] : K < \min_n \Xi. \tag{3}$$

*Proof.* Let $\overline{K} = \min_n \Xi - 1$. Then $\forall \, K \le \overline{K}$ is guaranteed that no correlation horizon $\xi$ is in the agent history $h_{t-K+1:t}$, hence the context length $K \le \min_n \Xi - 1$ generates the LTM problem exclusively. Since context length cannot be negative or zero, it turns out that $1 \le K \le \overline{K} = \min_n \Xi - 1$, which was required to prove. ∎

The following result, though intuitive, formalizes a practical criterion for isolating long-term memory evaluation by constraining the agent's context window. It serves as the foundation for configuring experiments in the Memory DM framework. According to Theorem 2, in a memory-intensive environment $\tilde{\mathcal{M}}_P$, the value of the context memory border $\overline{K}$ can be found as

$$\overline{K} = \min_n \Xi - 1 = \min_n \left\{ (t_r - t_e - \Delta t + 1)_n \right\}_n - 1. \tag{4}$$

Using Theorem 2, we can establish the necessary conditions for validating short-term memory: **1) Weak condition to validate short-term memory**: if $\overline{K} < K < \max_n \Xi$, then the memory-intensive

---

**Algorithm 1** Algorithm for setting up an experiment to test long-term and short-term memory in Memory DM framework.

---

**Require:** $\tilde{\mathcal{M}}_P$ – memory-intensive environment; $\mu(K)$ – memory mechanism.

**1. Estimate the number of $n$ event-recall pairs in the environment (Definition 4.6).**

     1. $n = 0 \rightarrow$ Environment is not suitable for testing long-term and short-term memory.

     2. $n \geq 1 \rightarrow$ Environment is suitable for testing long-term and short-term memory.

**2. Estimate context memory border $\overline{K}$ (4).**

     1. $\forall$ event-recall pair $(\beta(\alpha), \alpha)_i$ find corresponding $\xi_i, i \in [1..n]$.

     2. Determine $\overline{K}$ as $\overline{K} = \min \Xi - 1 = \min_n \{\xi_n\}_n - 1 = \min_n \left\{ (t_r - t_e - \Delta t + 1)_n \right\}_n - 1$

**3. Conduct an appropriate experiment (Definition 4.5).**

     1. To test short-term memory set $K > \overline{K}$.

     2. To test long-term memory set $K \leq \overline{K} \leq K_{eff} = \mu(K)$.

**4. Analyze the results.**

---

environment $\tilde{M}_P$ is used to validate both short-term and long-term memory. **2) Strong condition to validate short-term memory**: if $\max_n \Xi < K$, then the memory-intensive environment $\tilde{M}_P$ is used to validate exclusively short-term memory.

According to Theorem 2, if $K \in [1, \overline{K}]$, none of the correlation horizons $\xi$ will be in the agent's context, validating only long-term memory. When $\overline{K} < K < \max_n \Xi \leq T - 1$, long-term memory can still be tested, but some correlation horizons $\xi$ will fall within the agent's context and won't be used for long-term memory validation. In such a case it is not possible to estimate long-term memory explicitly. When $K \geq \max_n \Xi$, all correlation horizons $\xi$ are within the agent's context, validating only short-term memory. Summarizing the obtained results, the final division of the required agent context lengths for short-term memory and long-term memory validation is as follows: (i) $K \in [1, \overline{K}] \Rightarrow$ **validating LTM only**; (ii) $K \in (\overline{K}, \max_n \Xi) \Rightarrow$ **validating both STM and LTM**; (iii) $K \in [\max_n \Xi, \infty) \Rightarrow$ **validating STM only**.

## 4.2 LONG-TERM MEMORY IN MEMORY DM

As defined in Definition 4.5, short-term Memory DM tasks arise when event-recall pairs in $\tilde{\mathcal{M}}_P$ fall within the agent's context ($\xi \leq K$), allowing decisions based on local correlations. This holds regardless of how large $K$ is. Examples include (Esslinger et al., 2022; Grigsby et al., 2024; Ni et al., 2023). Validating STM is simple: increase $K$. In contrast, testing long-term memory requires more care and is typically more informative.

Memory DM tasks requiring long-term memory occur when event-recall pairs in the memory-intensive environment $\tilde{\mathcal{M}}_P$ are outside the agent's context ($\xi > K$). In this case, memory involves the agent's ability to connect information beyond its context, necessitating memory mechanisms (Definition 4.7) that can manage interaction histories $h$ longer than the agent's base model can handle.

**Definition 4.7** (**Memory mechanisms**). *Let the agent process histories $h_{t-K+1:t}$ of length $K$ at the current time $t$, where $K \in \mathbb{N}$ is agents context length. Then, a **memory mechanism** $\mu(K) : \mathbb{N} \to \mathbb{N}$ is defined as a function that, for a fixed $K$, allows the agent to process sequences of length $K_{eff} \geq K$, i.e., to establish global correlations out of context, where $K_{eff}$ is the effective context.*

$$\mu(K) = K_{eff} \geq K. \tag{5}$$

*Memory mechanisms are key to LTM tasks by recalling out-of-context information in Memory DM.*

**Example of memory mechanism.** Consider an agent based on an RNN architecture that can process $K = 1$ triplets of tokens $(o_t, a_t, r_t)$ at all times $t$. By using memory mechanisms $\mu(K)$ (e.g., as in Hausknecht & Stone (2015)), the agent can increase the number of tokens processed in a single step without expanding the context size of its RNN architecture. Therefore, if initially in a memory-intensive environment $\tilde{\mathcal{M}}_P : \xi > K = 1$, it can now be represented as $\tilde{\mathcal{M}}_P : \xi \leq K_{eff} = \mu(K)$. Here, the memory mechanism $\mu(K)$ refers to the RNNs recurrent updates to its hidden state.

Thus, validating an agent's ability to solve long-term memory problems in the Memory DM framework reduces to validating the agent's memory mechanisms $\mu(K)$. **To design correct experiments in such a case, the following condition must be met:**

$$\tilde{\mathcal{M}}_P : K \le \overline{K} < \xi \le K_{eff} = \mu(K) \tag{6}$$

According to our definitions, agents with memory mechanisms in the Memory DM framework that solve LTM tasks can also handle STM tasks, but not vice versa. The algorithm for setting up experiments to test an agent's STM or LTM is outlined in Algorithm 1.

### 4.3 EXAMPLE OF $\Xi$ AND $\xi$ ESTIMATES

Following the proposed methodology (Algorithm 1), we estimated the sets of correlation horizons $\Xi$ and minimal recall distances $\xi$ for a range of popular memory-intensive tasks (Table Table 2), including *Passive T-Maze* (Ni et al., 2023), *Minigrid-Memory* (Chevalier-Boisvert et al., 2023), *ViZDoom-Two-Colors* (Sorokin et al., 2022), *Memory Maze* (Pasukonis et al., 2022), *Memory Cards* (Esslinger et al., 2022), *Mortar Mayhem* and *Mystery Path* (Pleines et al., 2025), *POP-Gym–Autoencode* and *POPGym-RepeatPrevious* (Morad et al., 2023a).

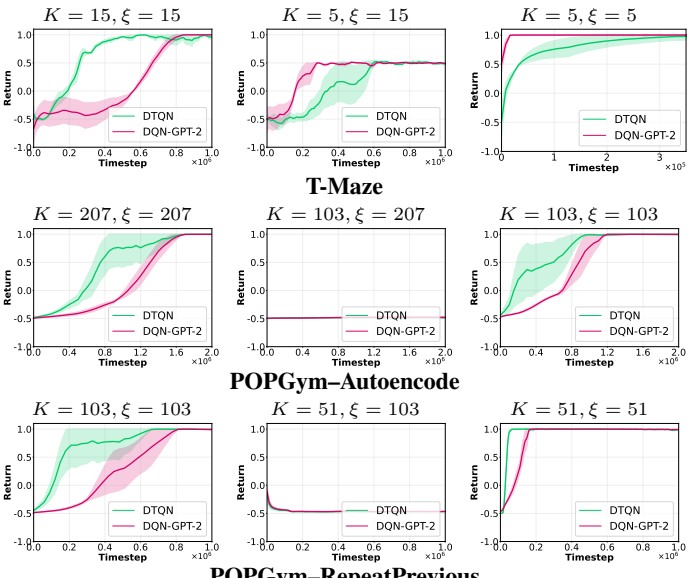

Figure 4: Performance of Online RL agents "with memory" across different memory configurations. Each row shows a specific environment: T-Maze, POPGym-Autoencode, and POPGym-RepeatPrevious, with varying agent context length $K$ and correlation horizons $\xi$. The STM $\leftrightarrows$ LTM transitions reflect the relative nature of the settings to test memory, depending on both agent and environment parameters.

**Example: Testing Memory in Passive T-Maze** In Passive T-Maze, the agent sees a cue at the start of a corridor and must turn correctly at the junction. The episode lasts $T = L+1$, where $L$ is the corridor length. Using Algorithm 1: **1)** There's one event-recall pair ($n = 1$), so the task suits both STM and LTM. **2)** The event lasts one step ($\Delta t = 0$), so $\xi = T$, and $\overline{K} = T - 1$. **3)** Varying $T$ or context size $K$ lets us test STM (if $K > \overline{K}$) or LTM (if $K \le \overline{K} \le \mu(K)$). While $K = \overline{K}$ is enough in theory, choosing smaller $K$ better reveals memory mechanism effects.

## 5 EXPERIMENTS

We evaluate memory-enhanced RL agents using the Memory DM framework to distinguish short- vs. long-term memory. Our experiments stress the need for proper methodology (Algorithm 1) and show how poor setups can misrepresent memory use. We test four memory-intensive tasks: Passive T-Maze and Minigrid-Memory (cue recall), and POPGym–Autoencode and RepeatPrevious (observation reconstruction and action repeat), all requiring recall over time. In the online setting, we evaluate DTQN (Esslinger et al., 2022), DQN-GPT-2, and SAC-GPT-2 (Ni et al., 2023) with attention-based memory. Offline, we test DT (Chen et al., 2021) and BC-LSTM to compare attention vs. recurrence. In all cases, we vary agent context $K$ and task horizon $\xi$ to isolate memory types and reveal model limitations.

## 5.1 PITFALLS OF NAIVE MEMORY TESTS

Proper evaluation of memory in RL agents requires distinguishing STM from LTM by accounting for correlation horizons $\xi$. Without this, STM and LTM effects blur, misrepresenting agent capacity. We illustrate this with SAC-GPT-2 in Minigrid-Memory under two setups: (i) fixed $L = 21$ ($\xi = 22$), and (ii) variable $L$ ($\xi \in [7, 22]$), testing STM ($K = 22$) and LTM ($K = 14$) settings. As shown in Figure 5, the *variable* setup gives high success for both settings, implying good memory.

But in the *fixed* case, LTM fails, revealing the agent's true limit. Mixed-horizon tasks can hide LTM deficits - only fixed $\xi > K$ setups expose them. Proper LTM evaluation requires controlling the correlation horizon $\xi$ relative to the agent's context $K$. Without this, STM effects may dominate and misrepresent the agent's memory type. Our methodology provides a principled way to avoid this confusion.

Table 2: Correlation horizons $\xi$ and LTM thresholds $K$ for popular memory-intensive tasks. $L$ is corridor length, $T$ is episode length. (f) and (v) denote fixed and variable setups. POPGym entries show values for the easy setting; for easy / medium / hard, $\Xi$ becomes $\{2, 4, \ldots, 104/208/312\}$ for Autoencode and $\{5/33/65\}$ for RepeatPrevious.

| Task | $\Xi$ | $\xi$ | LTM if $K <$ |
|---|---|---|---|
| Passive T-Maze | $\{L+1\}$ | $L+1$ | $L+1$ |
| Minigrid-Memory (f) | $\{L+1\}$ | $L+1$ | $L+1$ |
| Minigrid-Memory (v) | $[7, L+1]$ | 7 | 7 |
| ViZDoom-Two-Colors | $[2, 2055]$ | 2 | 2 |
| Memory Maze 9x9 | $[28, 1000]$ | 28 | 28 |
| Memory Maze 15x15 | $[45, 4000]$ | 45 | 45 |
| Memory Cards | $[2, T]$ | 2 | 2 |
| Mortar Mayhem (finite) | $[38, 218]$ | 38 | 38 |
| Mystery Path (finite) | $[8, 26]$ | 8 | 8 |
| POPGym–Autoencode | $[2, 104]$ | 2 | 2 |
| POPGym–RepeatPrevious | $\{5\}$ | 5 | 5 |

## 5.2 THE RELATIVE NATURE OF AN AGENT'S MEMORY

According to Algorithm 1, the experimental setup for testing agent memory types (LTM and STM) depends on two parameters: the agent's context length $K$ and the context memory border $\overline{K}$, which in turn is determined by the environment's correlation horizon $\xi$. Verifying LTM or STM requires adjusting $K$ or $\xi$ while keeping the other fixed. This section outlines how these parameters interact in memory testing. An agent's memory cannot be defined in isolation – it arises from the interplay between its context $K$ and the environment's horizons $\xi$. Thus, the same agent may exhibit either STM or LTM behavior depending on the task setup.

We test DTQN and DQN-GPT-2 in three memory-intensive tasks: Passive T-Maze, POPGym Autoencode, and RepeatPrevious – by varying $K$ and $\xi$ to simulate STM ($\xi \le K$) and LTM ($\xi > K$); as shown in Figure 4, performance is high when $\xi \le K$ but drops sharply for $\xi > K$, confirming that long-range dependencies require explicit memory mechanisms. These results confirm memory is relative: LTM depends on both temporal distance and agent design. Without controlling $K$ and $\xi$, memory claims are unreliable. Our $K$-$\xi$ framework ensures consistent, interpretable evaluation.

## 5.3 GENERALIZATION ACROSS SEQUENCE LENGTHS

Evaluating memory in RL agents requires distinguishing true long-term memory (LTM) from memorization within fixed context. To demonstrate this, we test DT and BC-LSTM on T-Maze: both are trained on specific corridor lengths and evaluated on both seen and longer, unseen ones. This setup tests whether agents can recall cue information beyond training range - an LTM indicator. Figure 6 shows success heatmaps across training and validation lengths. The diagonal indicates in-distribution performance; extrapolation lies to the right.

While both models process sequences and are labeled as memory-enhanced, our framework reveals key differences. DT relies on a fixed attention window and operates with short-term memory, while LSTM uses a recurrent state, enabling true LTM. T-Maze results expose this gap: DT performs well when validation lengths

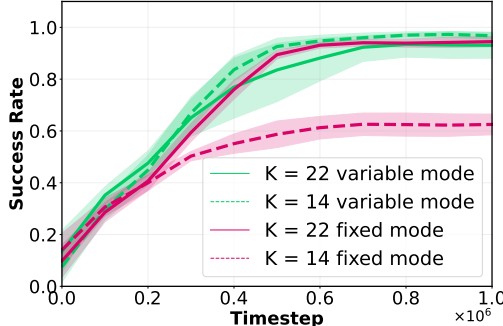

Figure 5: SAC-GPT-2 in Minigrid-Memory ($L = 21$) with short- ($K = 22$) and long-term ($K = 14$) memory setups. Variable mode (green) masks memory limits; fixed mode (red) reveals failure at $K = 14$, demonstrating lack of long-term memory – made evident by our evaluation method.

stay within context but fails for $L > 90$, whereas BC-LSTM generalizes to much longer sequences, demonstrating effective LTM. If evaluated only on shorter validation lengths, DT may appear stronger, masking memory limitations. Both perform well for lengths $\leq 150$, but at training length 300, DT scores $100\%$, while BC-LSTM drops to $0.87$. For longer training (600, 900), BC-LSTM collapses, DT remains high – misleadingly favoring STM.

DT succeeds only when the correlation horizon satisfies $\xi \leq K$, but fails whenever $\xi > K$, which demonstrates that DT lacks LTM. In contrast, BC-LSTM, on the horizons where it trains reliably, can exploit dependencies beyond the training range and therefore exhibits LTM within those regimes. Our framework makes this distinction precise by separating architectural limits from memory function, ensuring that DT is correctly identified as an STM agent, while BC-LSTM is capable of LTM on the ranges where it learns effectively (i.e. without vanishing gradients problem (Trinh et al., 2018)).

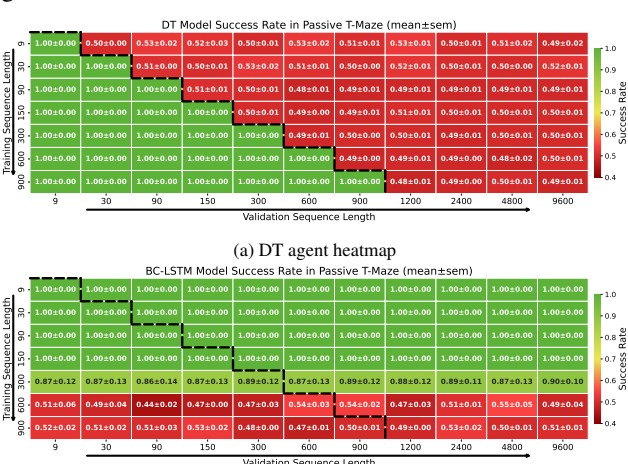

(a) DT agent heatmap

(b) BC-LSTM agent heatmap

Figure 6: Generalization on Passive T-Maze. Each heatmap shows success rates for (a) DT and (b) BC-LSTM across training (vertical) and validation (horizontal) sequence lengths. DT succeeds only when validation $\leq$ training, showing short-term memory limits. BC-LSTM generalizes beyond training, indicating strong long-term memory.

## 6 CONCLUSION

We propose a unified framework for classifying and evaluating memory in RL agents, grounded in neuroscience-inspired definitions of short- and long-term declarative memory. By introducing the concept of correlation horizon and formalizing memory-intensive environments, we enable precise evaluation of agent memory. Our methodology reveals key differences between architectures: transformers like DTQN or DT rely mainly on short-term memory, while recurrent models such as BC-LSTM exhibit long-term memory. Experiments on T-Maze, MiniGrid, and POPGym confirm the need for proper setups to avoid misleading conclusions. The framework clarifies how memory mechanisms shape behavior and could be extended to include additional systems from cognitive science, such as working or episodic memory, and to explore whether new types emerge in complex RL tasks.

As a direction for future work, it would be valuable to study adaptive dynamic updating of memory representations, since most existing work focuses primarily on memorization and retention rather than on how agents revise stored information over time.

## REPRODUCIBILITY STATEMENT

We have taken several measures to ensure the reproducibility of our results. **Model details:** The formalization of our framework – covering Memory DM, STM/LTM, and the correlation horizon – is provided in Section 4, with precise definitions in Definition 4.1, Definition 4.2, and Definition 4.5, and the experimental configuration procedure in Algorithm 1. **Theoretical results:** Assumptions and complete statements (including the definition of memory-intensive environments) are given in Definition 4.6, and key results with proofs appear in Theorem 2 and its accompanying discussion. **Experimental setup:** Tasks, training procedures, and evaluation protocols are reported in Section 5, with validation protocol details in Section F.1, hyperparameters in Section F.1, and environment descriptions in Appendix F. **Baselines:** Baseline selections and configurations are documented in Section 5, with their hyperparameters listed in Section F.1. **Code and data:** An anonymous repository containing source code, training scripts, and configuration files submitted as supplementary material. Together, these resources allow for full replication of our theoretical analyses and empirical results.

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

# A    APPENDIX – GLOSSARY

In this section, we provide a comprehensive glossary of key terms and concepts used throughout this paper. The definitions are intended to clarify the terminology proposed in our research and to ensure that readers have a clear understanding of the main elements underpinning our work.

1. $\mathcal{M}$ – MDP environment
2. $\mathcal{M}_P$ – POMDP environment
3. $\tilde{\mathcal{M}}_P$ – memory-intensive environment
4. $h_{0:t-1} = \{(o_i, a_i, r_i)\}_{i=0}^{t-1}$ – agent history of interactions with environment
5. $K$ – agent base model context length
6. $\overline{K}$ – context memory border of the agent, such that $K \in [1, \overline{K}] \Leftrightarrow$ strictly LTM problem
7. $\mu(K)$ – memory mechanism that increases number of steps available to the agent to process
8. $K_{eff} = \mu(K)$ – the agent effective context after applying the memory mechanism
9. $\alpha_{t_e}^{\Delta t} = \{(o_i, a_i, r_i)\}_{i=t_e}^{t_e+\Delta t}$ – an event starting at time $t_e$ and lasting $\Delta t$, which the agent should recall when making a decision in the future
10. $\beta_{t_r} = \beta_{t_r}(\alpha_{t_e}^{\Delta t}) = a_t \mid (o_t, \alpha_{t_e}^{\Delta t})$ – the moment of decision making at time $t_r$ according to the event $\alpha_{t_e}^{\Delta t}$
11. $\xi = t_r - t_a - \Delta t + 1$ – an event's correlation horizon

# B    APPENDIX – ADDITIONAL NOTES ON THE MOTIVATION FOR THE ARTICLE

## B.1    WHY USE DEFINITIONS FROM NEUROSCIENCE?

Definitions from neuroscience and cognitive science, such as short-term and long-term memory, as well as declarative and procedural memory, are already well-established in the RL community, but do not have common meanings and are interpreted in different ways. We strictly formalize these definitions to avoid possible confusion that may arise when introducing new concepts and redefine them with clear, quantitative meanings to specify the type of agent memory, since the performance of many algorithms depends on their type of memory.

In focusing exclusively on memory within RL, we do not attempt to exhaustively replicate the full spectrum of human memory. Instead, our goal is to leverage the intuitive understanding of neuroscience concepts already familiar to RL researchers. This approach avoids the unnecessary introduction of new terminology into the already complex Memory RL domain. By refining and aligning existing definitions, we create a robust framework that facilitates clear communication, rigorous evaluation, and practical application in RL research.

## B.2    ON PRACTICAL APPLICATIONS OF OUR FRAMEWORK

The primary goal of our framework is to address practical challenges in RL by providing a robust classification of memory types based on temporal dependencies and the nature of memorized information. This classification is essential for standardizing memory testing and ensuring that RL agents are evaluated under conditions that accurately reflect their capabilities.

In RL, memory is interpreted in various ways, such as transformers with large context windows, recurrent networks, or models capable of skill transfer across tasks. However, these approaches often vary fundamentally in design, making comparisons unreliable and leading to inconsistencies in testing. Our framework resolves this by providing a clear structure to evaluate memory mechanisms under uniform and practical conditions.

The proposed definitions of declarative and procedural memory use two straightforward numerical parameters: the number of environments ($n_{envs}$) and episodes ($n_{eps}$). These parameters allow researchers to reliably determine the type of memory required for a task. This simplicity and alignment with numerical parameters make the framework practical and widely applicable across diverse RL problems.

Moreover, the division of declarative memory into long-term and short-term memory, as well as the need to use a balance between the agent's context length $K$ and the correlation horizons of the environment $\xi$ when conducting the experiment, allows us to unambiguously determine which type of memory is present in the agent. This clarity ensures fair comparisons between agents with similar memory mechanisms and highlights specific limitations in an agent's design. By aligning memory definitions with practical testing requirements, the framework provides actionable insights to guide the development of memory-enhanced RL agents.

## C   APPENDIX – MEMORY MECHANISMS

In RL, memory has several meanings, each of which is related to a specific class of different tasks. To solve these tasks, the authors use various memory mechanisms. The most prevalent approach to incorporating memory into an agent is through the use of Recurrent Neural Networks (RNNs) (Rumelhart et al., 1986), which are capable of handling sequential dependencies by maintaining a hidden state that captures information about previous time steps (Wierstra et al., 2010; Hausknecht & Stone, 2015; Sorokin et al., 2015; Duan et al., 2016; Song et al., 2018; Zintgraf et al., 2020) (pure LTM, according to our taxonomy). Another popular way to implement memory is to use Transformers (Vaswani et al., 2017), which use self-attention mechanisms to capture dependencies inside the context window (Parisotto et al., 2020; Lampinen et al., 2021; Esslinger et al., 2022; Melo, 2022; Team et al., 2023; Pramanik et al., 2023; Robine et al., 2023; Ni et al., 2023; Grigsby et al., 2024; Shala et al., 2024) (STM in case of classical transformers without additional memory mechanisms or LTM if we use recurrent memory, activation caching, etc.). State-space models (SSMs) (Gu et al., 2021; Smith et al., 2023; Gu & Dao, 2023) combine the strengths of RNNs and Transformers and can also serve to implement memory through preservation of system state (Hafner et al., 2019; Lu et al., 2023; Becker et al., 2024; Samsami et al., 2024) (LTM, according to our taxonomy). Temporal convolutions may be regarded as an effective memory mechanism, whereby information is stored implicitly through the application of learnable filters across the time axis (YuXuan Liu & Hsieh, 2016; Mishra et al., 2018) (STM, since memory is represented as a fixed-size temporal convolution, analogous to an attention window). A world model (Ha & Schmidhuber, 2018) which builds an internal environment representation can also be considered as a form of memory. One method for organizing this internal representation is through the use of a graph, where nodes represent observations within the environment and edges represent actions (Morad et al., 2021; Zhu et al., 2023; Kang et al., 2024b).

A distinct natural realization of memory is the utilization of an external memory buffer, which enables the agent to retrieve pertinent information. This approach can be classified into two categories: read-only (writeless) (Oh et al., 2016; Lampinen et al., 2021; Goyal et al., 2022; Cherepanov et al., 2024) and read/write access (Graves et al., 2016; Zaremba & Sutskever, 2016; Parisotto & Salakhutdinov, 2017a).

Memory can also be implemented without architectural mechanisms, relying instead on agent policy. For instance, in the work of Deverett et al. (2019), the agent learns to encode temporal intervals by generating specific action patterns. This approach allows the agent to implicitly represent timing information within its behavior, showcasing that memory can emerge as a result of policy adaptations rather than being explicitly embedded in the underlying neural architecture.

Using these memory mechanisms, both decision-making tasks based on information from the past within a single episode and tasks of fast adaptation to new tasks are solved. However, even in works using the same underlying base architectures to solve the same class of problems, the concepts of memory may differ.

## D   APPENDIX – POMDP

### D.1   POMDP

The Partially Observable Markov Decision Process (POMDP) is a generalization of the Markov Decision Process (MDP) that models sequential decision-making problems where the agent has incomplete information about the environment's state. POMDP can be represented as a tuple $\mathcal{M}_P = \langle \mathcal{S}, \mathcal{A}, \mathcal{O}, \mathcal{P}, \mathcal{R}, \mathcal{Z} \rangle$, where $\mathcal{S}$ denotes the set of states, $\mathcal{A}$ is the set of actions, $\mathcal{O}$ is the set of observations and $\mathcal{Z} = \mathcal{P}(o_{t+1} \mid s_{t+1}, a_t)$ is an observation function such that $o_{t+1} \sim \mathcal{Z}(s_{t+1}, a_t)$.

An agent takes an action $a_t \in \mathcal{A}$ based on the observed history $h_{0:t-1} = \{(o_i, a_i, r_i)\}_{i=0}^{t-1}$ and receives a reward $r_t = \mathcal{R}(s_t, a_t)$. It is important to note that state $s_t$ is not available to the agent at time $t$. In the case of POMDPs, a policy is a function $\pi(a_t \mid o_t, h_{0:t-1})$ that uses the agent history $h_{0:t-1}$ to obtain the probability of the action $a_t$. Thus, in order to operate effectively in a POMDPs, an agent must have memory mechanisms to retrieve a history $h_{0:t-1}$. Partial observability arises in a variety of real-world situations, including robotic navigation and manipulation tasks, autonomous vehicle tasks, and complex decision-making problems.

## E    APPENDIX – META REINFORCEMENT LEARNING

In this section, we explore the concept of Meta-Reinforcement Learning (Meta-RL), a specialized domain within POMDPs that focuses on equipping agents with the ability to learn from their past experiences across multiple tasks. This capability is particularly crucial in dynamic environments where agents must adapt quickly to new challenges. By recognizing and memorizing common patterns and structures from previous interactions, agents can enhance their efficiency and effectiveness when facing unseen tasks.

Meta-RL is characterized by the principle of "*learning to learn*", where agents are trained not only to excel at specific tasks but also to generalize their knowledge and rapidly adjust to new tasks with minimal additional training. This adaptability is achieved through a structured approach that involves mapping data collected from various tasks to policies that guide the agent's behavior.

Meta-RL algorithm is a function $f_\theta$ parameterized with *meta-parameters* that maps the data $\mathcal{D}$, obtained during the process of training of RL agent in MDPs (tasks) $\mathcal{M}_i \sim p(\mathcal{M})$, to a policy $\pi_\phi : \phi = f_\theta(\mathcal{D})$. The process of learning the function $f$ is typically referred to as the *outer-loop*, while the resulting function f is called the *inner-loop*. In this context, the parameters $\theta$ are associated with the outer-loop, while the parameters $\phi$ are associated with the inner-loop. Meta-training proceeds by sampling a task from the task distribution, running the inner-loop on it, and optimizing the inner-loop to improve the policies it produces. The interaction of the inner-loop with the task, during which the adaptation happens, is called a *lifetime* or a *trial*. In Meta-RL, it is common for $\mathcal{S}$ and $\mathcal{A}$ to be shared between all of the tasks and the tasks to only differ in the reward $\mathcal{R}(s, a)$ function, the dynamics $\mathcal{P}(s' \mid s, a)$, and initial state distributions $P_0(s_0)$ (Beck et al., 2024).

## F    APPENDIX – EXPERIMENT DETAILS

This section provides an extended description of the environments used in this work.

**Passive-T-Maze (Ni et al., 2023).**    In this T-shaped maze environment, the agent's goal is to move from the starting point to the junction and make the correct turn based on an initial signal. The agent can select from four possible actions: $a \in left, up, right, down$. The signal, denoted by the variable $clue$, is provided only at the beginning of the trajectory and indicates whether the agent should turn up ($clue = 1$) or down ($clue = -1$). The episode duration is constrained to $T = L + 1$, where $L$ is the length of the corridor leading to the junction, which adds complexity to the task. To facilitate navigation, a binary variable called $flag$ is included in the observation vector. This variable equals 1 one step before reaching the junction and 0 at all other times, indicating the agent's proximity to the junction. Additionally, a noise channel introduces random integer values from the set $-1, 0, +1$ into the observation vector, further complicating the task. The observation vector is defined as $o = [y, clue, flag, noise]$, where $y$ represents the vertical coordinate.

The agent receives a reward only at the end of the episode, which depends on whether it makes a correct turn at the junction. A correct turn yields a reward of $1$, while an incorrect turn results in a reward of $0$. This configuration differs from the conventional Passive T-Maze environment (Ni et al., 2023) by featuring distinct observations and reward structures, thereby presenting a more intricate set of conditions for the agent to navigate and learn within a defined time constraint. To transition from a sparse reward function to a dense reward function, the environment is parameterized using a penalty defined as $penalty = -\frac{1}{T-1}$, which imposes a penalty on the agent for each step taken within the environment. Thus, this environment has a 1D vector space of observations, a discrete action space, and sparse and dense configurations of the reward function.

**Minigrid-Memory** (Chevalier-Boisvert et al., 2023). Minigrid-Memory is a two-dimensional grid-based environment specifically crafted to evaluate an agent's long-term memory and credit assignment capabilities. The layout consists of a T-shaped maze featuring a small room at the corridor's outset, which contains an object. The agent is instantiated at a random position within the corridor. Its objective is to navigate to the chamber, observe and memorize the object, then proceed to the junction at the maze's terminus and turn towards the direction where the object, identical to that in the initial chamber, is situated. A reward function defined as $r = 1 - 0.9 \times \frac{t}{T}$ is awarded upon successful completion, while failure results in a reward of zero. The episode concludes when the agent either makes a turn at a junction or exhausts a predefined time limit of 95 steps. To implement partial observability, observational constraints are imposed on the agent, limiting its view to a $3 \times 3$ frame size. Thus, this environment has a 2D space of image observations, a discrete action space, and sparse reward function.

### F.1 EXPERIMENTAL PROTOCOL

For each experiment, we conducted three runs of the agents with different initializations and performed validation during training using 100 random seeds ranging from 0 to 99. The results are presented as the mean success rate (or reward) ± the standard error of the mean (SEM).

Table 3: Online RL baselines hyperparameters used in the Minigrid-Memory and Passive T-Maze experiments.

Table 4: SAC-GPT-2

| Hyperparameter | Value |
| --- | --- |
| Number of layers | 2 |
| Number of attention heads | 2 |
| Hidden dimension | 256 |
| Batch size | 64 |
| Optimizer | Adam |
| Learning rate | 3e-4 |
| Dropout | 0.1 |
| Replay buffer size | 1e6 |
| Discount ($\gamma$) | 0.99 |
| Entropy temperature | 0.1 |

Table 5: DQN-GPT-2

| Hyperparameter | Value |
| --- | --- |
| Number of layers | 2 |
| Number of attention heads | 2 |
| Hidden dimension | 256 |
| Batch size | 64 |
| Optimizer | Adam |
| Learning rate | 3e-4 |
| Dropout | 0.1 |
| Replay buffer size | 1e6 |
| Discount ($\gamma$) | 0.99 |

Table 6: DTQN

| Hyperparameter | Value |
| --- | --- |
| Number of layers | 4 |
| Number of attention heads | 8 |
| Hidden dimension | 128 |
| Batch size | 32 |
| Optimizer | Adam |
| Learning rate | 3e-4 |
| Dropout | 0.1 |
| Replay buffer size | 5e5 |
| Discount ($\gamma$) | 0.99 |

Table 7: Offline RL baselines hyperparameters used for Decision Transformer and BC-LSTM in T-Maze experiments.

Table 8: Decision Transformer (DT)

| Hyperparameter | Value |
| --- | --- |
| Number of layers | 8 |
| Number of attention heads | 4 |
| Hidden dimension ($d_{\text{model}}$) | 128 |
| Feedforward dimension ($d_{\text{inner}}$) | 128 |
| Head dimension ($d_{\text{head}}$) | 128 |
| Context length ($K$) | $3T$ |
| Dropout | 0.0 |
| DropAttention | 0.0 |
| Optimizer | AdamW |
| Learning rate | 1e-4 |
| Weight decay | 0.1 |
| Adam betas | (0.9, 0.999) |
| Batch size | 64 |
| Warmup steps | 1000 |
| Epochs | 200 |

Table 9: BC-LSTM

| Hyperparameter | Value |
| --- | --- |
| Number of layers | 1 |
| Hidden dimension ($d_{\text{model}}$) | 64 |
| Bidirectional | False |
| Effective Context length ($K_{eff}$) | $3T$ |
| Dropout | 0.0 |
| Optimizer | AdamW |
| Learning rate | 3e-4 |
| Weight decay | 0.01 |
| Adam betas | (0.9, 0.999) |
| Batch size | 64 |
| Warmup steps | 100 |
| Epochs | 100 |

