# OpenReview forum: "Unraveling the Complexity of Memory in RL Agents: an Approach for Classification and Evaluation"
_ICLR.cc/2026/Conference — ICLR 2026 Poster_

### Official Review · Reviewer_Wrt1 · 2025-10-24

**Soundness:** 2
**Presentation:** 3
**Contribution:** 2
**Rating:** 6
**Confidence:** 3

**Summary:**

The overcome the limitation that the term “memory” in RL remains inconsistently defined and often misaligned with what experiments actually test, this paper introduces a unified taxonomy of memory in RL based on temporal dependencies and task structure. More specifically, they give a formal definition for the Memory Decision-Making with declarative long-term and short-term memory and Meta-RL with procedural memory. Then an experimental algorithm is proposed to test agent memory types based on context length and context memory border. Experiments on T-Maze, MiniGrid, and POPGym show that the proposed algorithm is efficient at distinguishing memory types.

**Strengths:**

1.	This paper is well-written and well-organized. It provides a clear taxonomy of memory in RL with clear definitions.
2.	The classification of long-term and short-term memory with context length and context memory border is reasonable.
3.	The experiments in Section 5.1 show that the variable correlation horizon could hide long-term memory’s deficits.

**Weaknesses:**

1.	The definition of memory in RL is not a novel thing. For example, long-term and short-term memory are already well-studied in the domain of RL. And skill transfer in Meta-RL is also a popular concept.
2.	Many works in memory RL such as [1,2] are not classified using the proposed experimental classification algorithm (Algorithm 1).
3.	There are some typos such as “According to Theorem Theorem 2,”.

References

[1] Beck, Jacob, et al. "Amrl: Aggregated memory for reinforcement learning." International Conference on Learning Representations. 2020.

[2] Loynd, Ricky, et al. "Working memory graphs." International conference on machine learning. PMLR, 2020.

**Questions:**

1.	Could the authors classify more classical works in the domain of memory RL?
2.	Could this method be applied to agentic RL with LLMs?
3.	Are there any future research directions based on this work?

---

> ### Author Response · Authors · 2025-11-22
> **Ofifcial comment by Authors**
>
> We thank Reviewer Wrt1 for their insightful comments on our work. We particularly appreciate the acknowledgment that the proposed taxonomy of memory in RL is clear, and that the experimental evidence successfully demonstrates how improper task setups can obscure the absence of long-term memory in existing agents.
>
> > W1. The definition of memory in RL is not a novel thing. For example, long-term and short-term memory are already well-studied in the domain of RL. And skill transfer in Meta-RL is also a popular concept.
>
> We respectfully disagree with the claim that the definition of memory in RL presented in the paper lacks novelty. While it is correct that terms such as “short-term”, “long-term”, “declarative”, and “procedural” memory appear in existing RL literature, their usage is informal, mutually inconsistent, and not grounded in operational criteria. Our contribution is not the invention of new terminology, but the formalization of previously vague concepts into mathematically testable objects.
>
> > W2. Many works in memory RL such as [1,2] are not classified using the proposed experimental classification algorithm (Algorithm 1).
> > W3. There are some typos such as “According to Theorem Theorem 2,”.
>
> We thank the Reviewer for pointing out these additional works on memory mechanisms. Both papers have been incorporated into the revised “Related Work” section. We also note that Appendix C (“Memory Mechanisms”) already discusses the main architectural families used to realize agent memory, including those based on recurrent state accumulation, key-value external memory, and graph-structured working memory, which cover the mechanisms proposed in the cited works.
>
>
> Algorithm 1 provides a model-agnostic procedure for isolating short-term and long-term memory regimes in any Memory-DM task. However, the quantities $K$ (agent context length) and $K_{eff}$ (effective context induced by the memory mechanism $\mu(K)$) are architecture-dependent parameters, and must therefore be estimated separately for each model. Our framework deliberately leaves this step open, since different architectures expand their effective context by fundamentally different mechanisms (e.g., recurrent state propagation, learned retrieval, graph message passing). The evaluation protocol remains valid for all such models once $K$ and $K_{eff}$ are specified.
>
> > Q1. Could the authors classify more classical works in the domain of memory RL?
>
> Thank you for your question. To answer it, we've expanded Appendix, Section C - "Memory Mechanisms," adding a classification of several classic works according to our taxonomy (see attached revised .pdf file).
>
> > Q2. Could this method be applied to agentic RL with LLMs?
>
> Thank you for the interesting question! First of all, I'd like to point out that we are not very familiar with the field of agentic RL with LLMs. However, if we can formulate the text generation process as an RL environment, as described by POMDP, then exactly the same problems associated with the limited context window of the transformer arise that we encounter in the memory RL domain. In this case, we can also introduce various memory mechanisms (see RMT [1] or TrXL [2] papers) to overcome the limited context for solving LTM problems.
>
> > Q3. Are there any future research directions based on this work?
>
> We think that another heavily underexplored area is the study of the memory capacity of memory-enhanced RL agents (how to define? how to evaluate?), as well as the capabilities to update/rewrite/forget memories, since most existing work focuses on memorization and retention rather than adaptive dynamic updating.
>
>
> We are once again grateful to the Reviewer for their insightful comments. We hope we were able to address all questions and comments, and we are happy to answer any further questions during the subsequent discussion.
>
> [1] Bulatov, Aydar, Yury Kuratov, and Mikhail Burtsev. "Recurrent memory transformer." Advances in Neural Information Processing Systems 35 (2022): 11079-11091.
>
> [2] Dai, Zihang, et al. "Transformer-xl: Attentive language models beyond a fixed-length context." Proceedings of the 57th annual meeting of the association for computational linguistics. 2019.

---

### Official Review · Reviewer_RqbQ · 2025-10-27

**Soundness:** 3
**Presentation:** 2
**Contribution:** 3
**Rating:** 4
**Confidence:** 2

**Summary:**

This paper proposes a standard "K-$\xi$" definition of different kinds of "memory" in RL Agents. The majority of the paper is dedicated to establishing and defining the proposed classification system. Afterward, it explains how to use this classification system to evaluate LTM and STM separately and shows examples of the misleading conclusions of combined evaluations.

**Strengths:**

1. They standardized the definitions of different types of memory, thereby providing a framework for fair evaluation of subsequent models.
2. The definitions and formalizations are careful and rigorous.

**Weaknesses:**

The paper spends too much time on the basic theoretical and general analyses, while the experiments and analyses conducted under this theoretical framework are rather limited. Although the authors include several validation experiments demonstrating the necessity and importance of defining and distinguishing different types of memory, they do not provide any insightful conclusions or analyses derived from evaluations based on this distinction. In other words, what I expected to see was either a newly designed evaluation benchmark or tasks built upon this theoretical distinction of memory, or a comparative analysis of how the latest models or methods perform across different types of memory. But neither is included in the paper.

Frankly speaking, I am not very familiar with the expected workload or contribution standards for theoretical analyses in the RL field. Therefore, I have set my confidence score relatively low. Overall, what I expected to see here was the proposal of a new theory accompanied by analyses of the limitations of current research or guidance for the design of new models and methods. However, this paper seems to have achieved only the first part. In addition, since it does not introduce any new concepts but rather provides standardized definitions for existing ones, I feel that the overall workload and contribution of this work are somewhat limited. Again, as I am not fully familiar with the typical expectations for such work in RL, I am open to discussion and would be willing to raise both the rating and confidence scores if convinced.

**Questions:**

1. The order of the figures and tables is disorganized.
2. You claim that LSTM is better than Transformers in terms of LTM. What's your opinion on the following facts?
    * The performance of LSTM falls low when the training sequence is larger than 300. Its low performance under both seen validation samples and longer, unseen validation samples seems to indicate that the LSTM's capacity to perform both LTM and STM tasks falls short when the training sequence becomes too long. This precisely reflects the gradient vanishing problem of LSTM and seems to contradict your claim that "BC-LSTM retains LTM despite challenges like vanishing gradient."
    * Current studies have designed many methods to introduce LTM into transformers, such as maintaining an external memory bank and summarizing previous steps. How well do these methods perform in mitigating the LTM issues inherent in Transformer models?

---

> ### Author Response · Authors · 2025-11-22
> **Ofifcial comment by Authors**
>
> We thank Reviewer RqbQ for their careful evaluation of our work. We particularly appreciate the acknowledgment that the proposed framework standardizes definitions of agent memory and enables fair evaluation of future models.
>
> We thank for the comments and for explicitly noting uncertainty regarding expectations for theoretical contributions in the RL. We believe the concerns arise from a mismatch between the expectations stated in the review and the intended scope of the paper. The critique assumes that the contribution should include a new benchmark or extensive empirical comparisons across memory types. However, the paper does not claim to introduce a benchmark. Instead, it identifies a foundational methodological problem in current memory evaluation: the absence of a unified taxonomy and a quantitative protocol for distinguishing different forms of memory in RL agents.
>
> The primary contribution of this work is conceptual. While it is correct that we do not introduce entirely new memory terminology, the key novelty lies in formal unification. Concepts such as long-term vs. short-term memory, or declarative vs. procedural memory, already appear in RL literature but are used in mutually incompatible ways, often referring to completely different mechanisms or temporal structures. Our contribution, to the best of our knowledge, is the first mathematically explicit framework that links memory type, correlation horizon, context length, and evaluation protocol in a way that allows determining whether an agent truly uses memory rather than simply exploiting context leakage. To our knowledge, no prior work provides such a taxonomy or derives the necessary and sufficient conditions under which a task evaluates short- or long-term memory.
>
> The theoretical results and empirical analysis address exactly the issues raised by the reviewer.
>
> 1. The paper formalizes memory-intensive environments and establishes the correlation-horizon boundary that separates short- and long-term memory regimes.
>
> 2. The experiments demonstrate that, when evaluated under correct correlation-distance constraints, several widely used models fail to exhibit long-term memory despite being described as “memory-enhanced”. Conversely, under mixed-horizon task configurations, the same models appear successful, revealing that current evaluation practice can lead to incorrect conclusions.
>
> 3. The framework directly implies architectural guidance: any model whose effective context cannot exceed its base window cannot possess long-term memory under our definitions. This follows immediately from the formalism.
>
> The experiments are therefore not intended to constitute a benchmark but to validate the theoretical claims by demonstrating the consequences of misaligned evaluation protocols. A benchmark built on top of our framework is a natural next step, but such a benchmark requires precisely the definitions and methodology introduced here. The contribution should thus be evaluated as a foundational restructuring of how memory is defined and tested in RL, not as a benchmark or model paper.
>
> We appreciate that the reviewer is open to increasing both score and confidence. We hope the above clarifications make it clear that:
>
> - The work does not restate known ideas; it formally unifies fragmented ones and resolves long-standing inconsistencies.
>
> - The theoretical results directly expose what current evaluation protocols fail to measure.
>
> - The empirical section is purposefully minimal and serves to demonstrate that incorrect memory evaluation produces misleading scientific conclusions.
>
> > Q1. The order of the figures and tables
>
> We have corrected this in the updated version of the text.
>
> > Q2. You claim that LSTM is better than Transformers in terms of LTM
>
> We were referring to the part of the figure where BC-LSTM learns well, meaning it doesn't experience problems with vanishing gradients. Thus, at the horizons on which it learns, it can transfer information beyond the horizons on which it was trained. Conversely, DT can't do this even at such horizons, confirming its connection with our definitions of STM and LTM. We have reformulated  this point in the text.
>
> Regarding the second part of the question about Transformers with LTM mechanisms, there are indeed models that cope well with the LTM problems inherent in Transformer models (e.g., GTrXL [1] or RATE [2]). However, when considering history, we specifically consider “recurrence-only” (BC-LSTM) and “attention-only” (Decision Transformer) models to isolate the effects of “pure LTM” and “pure STM” memory mechanisms.
>
> We are once again grateful to the Reviewer RqbQ for their detailed comments on our work. We believe the revisions directly address all raised points and further clarify the core contributions. We are happy to engage in any additional discussion should it support the final assessment.
>
> [1] Parisotto, Emilio, et al. GTrXL, ICML 2020.
>
> [2] Cherepanov, Egor, et al. RATE, arXiv:2306.09459 2023.

---

> > ### Comment · Reviewer_RqbQ · 2025-11-27
> >
> > Dear Authors,
> >
> > Thank you for your kind reply and explanation. First, I would like to clarify that I am aware that the paper does not claim to introduce a benchmark. However, that is exactly why I think your paper is incomplete or insufficient as a standalone research work.
> >
> > More specifically, the main contribution of your work is the introduction of a clearer and more rigorous categorization of memory, which is indeed interesting and insightful. However, my primary concern is that proposing a new concept is not sufficient to support a standalone research work. In my view, a concept is only the starting point of a project; building on that concept, substantial additional work is needed to provide meaningful contributions to the field.
> >
> > To illustrate this, I gave two examples in my original review. One possibility is to use the new concept to guide the design of a new benchmark or dataset that can better evaluate or enhance certain dimensions of model performance. Another possibility is to conduct a thorough analysis and comparison of existing work based on the new concept, thereby providing conclusions that offer substantial guidance for model design or training. Although you have conducted some experiments, the scale of the experiments is quite limited, and the models involved are not among the more advanced and contemporary models or systems. As a result, the experiments do not present conclusions that are sufficiently valuable or instructive to readers.
> >
> > Therefore, in its current form, the work appears to have achieved only the proposal of a concept, without providing substantial contributions or meaningful insights. I believe you may persuade me in two ways: one is by adding corresponding work to further strengthen the paper, and the other is by convincing me that the conceptual contribution alone is sufficient to stand as an independent research work. If you are able to address this main concern, I would be happy to raise my score. However, given the current state of the work, I will temporarily maintain my original score.

---

> > > ### Author Response · Authors · 2025-12-02
> > >
> > > We sincerely thank the Reviewer for the continued engagement and for articulating the core concern clearly. We acknowledge that, in light of recent news, further bilateral discussion is no longer possible. Nevertheless, we provide clarifications on the remaining points below.
> > >
> > > The Reviewer argues that a conceptual contribution alone is insufficient as a standalone piece of research unless accompanied by a new benchmark or extensive empirical comparisons. **We respectfully disagree** and clarify why the primary purpose and **contribution of this work are both complete and impactful within the established norms of theoretical and methodological research in RL.**
> > >
> > > 1. **The main purpose of the work is to identify and formalize a foundational methodological problem**
> > >
> > > A central motivation of the paper is that the RL community currently has no principled, quantitative definition of memory, and as a consequence, widely used evaluations produce systematically misleading conclusions. This problem is documented both in the literature and empirically in our paper (e.g., ambiguous use of “long-term memory”, inability to distinguish STM vs LTM when $\xi$ varies across episodes, and inconsistent attribution of memory to architectural features).
> > >
> > > The contribution is therefore not merely a “new concept”; it is a **resolution of a long-standing conceptual gap that directly affects empirical practice.** The clarity of definitions (Definitions 4.4 - 4.6), the formal separation between STM/LTM via $\xi$ and $K$, and the identification of memory-intensive environments resolve inconsistencies that have accumulated for years in work on memory in RL. This type of theoretical normalization is well within the scope of standalone methodological contributions in RL.
> > >
> > > 2. **The paper does not only propose a definition - it proposes a practical resolution and a protocol for the community**
> > >
> > > Beyond clarifying terminology, the paper provides a concrete operational solution:
> > > - the correlation-horizon criterion
> > > - the notion of context memory border $K$
> > > - the formal definition of memory-intensive environments
> > > - a complete evaluation algorithm (Algorithm 1) that explicitly constructs LTM-only regimes
> > >
> > > These results transform informal ideas into a measurable protocol (Table 2). This is precisely the kind of contribution that stands independently.
> > >
> > > 3. **A new benchmark is not required - indeed, introducing one would contradict the purpose of the work**
> > >
> > > The Reviewer suggests adding a benchmark or dataset. We respectfully emphasize that this is **opposite** to the goal of the paper.
> > > Our formalism demonstrates that the real problem is not the lack of environments, but the **incorrect use of the existing ones.** As shown in Table 2 and in Section 5:
> > >
> > > - the same environment (T-Maze, MiniGrid-Memory, POPGym-Autoencode, etc.)
> > > - **can** evaluate only STM or only LTM
> > > - depending on how one configures $\xi$ relative to the agent’s context $K$
> > >
> > >
> > > Thus, the point is not to add another benchmark, but to **propose a method for the community on how to use existing benchmarks correctly to evaluate memory**. Algorithm 1 and Theorem 2 together provide a mathematically rigorous and practically applicable procedure for doing exactly this.
> > > A new benchmark would simply obscure the contribution, because the issue is conceptual and methodological, not a lack of environments.
> > >
> > > 4. **The experiments intentionally focus on isolating the methodological effect, not on breadth of models**
> > >
> > > The Reviewer notes that the empirical section does not include the newest or largest architectures. This is deliberate. The goal of the experiments is not to optimize performance or to start a leaderboard but to **validate the theoretical claim**: “If $\xi$ is not controlled relative to $K$, evaluations systematically overestimate an agent’s memory.
> > > This is demonstrated clearly in Fig. 4 (STM vs LTM transitions), Fig. 5 (variable $\xi$ hides LTM deficits), Fig. 6 (LSTM exhibits true LTM within the horizons it can stably learn; DT fails even there). Adding larger or more modern architectures does not change or strengthen the theoretical argument. What matters is showing the methodological failure mode, which the experiments achieve.

---

> > > > ### Author Response · Authors · 2025-12-02
> > > >
> > > > 5. **Why the conceptual contribution *is sufficient* as a standalone work**
> > > >
> > > > To summarize:
> > > >
> > > > 1. We diagnose a pervasive methodological flaw affecting a wide field of RL research
> > > > 2. We introduce a formal taxonomy that resolves long-standing conceptual inconsistencies.
> > > > 3. We derive measurable constructs that define when STM or LTM is required.
> > > > 4. We provide a complete evaluation protocol (Algorithm 1).
> > > > 5. We demonstrate that current agents are routinely mis-evaluated unless this protocol is followed.
> > > > 6. We show how widely used benchmarks already support rigorous STM/LTM isolation when used correctly.
> > > >
> > > >
> > > > This combination of theoretical unification, operational methodology, general applicability, and empirical validation of a methodological failure mode constitutes a complete and substantive standalone contribution.
> > > >
> > > > We again thank the Reviewer for the opportunity to further clarify the scope and significance of the work. We hope this explanation makes the completeness and impact of the contribution clear.

---

### Official Review · Reviewer_vvf9 · 2025-11-01

**Soundness:** 2
**Presentation:** 2
**Contribution:** 2
**Rating:** 4
**Confidence:** 2

**Summary:**

This paper proposes a clear, systematic, and practically meaningful framework for classifying and evaluating “memory” capabilities in reinforcement learning (RL) agents. Drawing inspiration from cognitive science—particularly classic memory typologies such as short-term vs. long-term memory and declarative vs. procedural memory—the authors formalize these concepts into operational definitions suitable for RL settings. The paper further introduces key notions such as “memory-intensive environments,” “correlation horizon (ξ),” and “context memory border (K),” and builds upon them a rigorous experimental methodology.

**Strengths:**

**Conceptual Clarification and Formalization**
The paper successfully translates ambiguous cognitive science terms (e.g., STM/LTM, declarative/procedural memory) into precise, quantifiable, and verifiable definitions within RL (see Definitions 4.4–4.6). This formalization fills a significant void in current RL literature, where “memory” is often used loosely or inconsistently.

**Proposal of a Unified Evaluation Framework**
The distinction between Memory Decision-Making (Memory DM) and Meta-RL is clearly articulated. The operational definitions of declarative memory (nenvs × neps = 1) and procedural memory (nenvs × neps > 1) provide a practical and measurable criterion for categorizing memory types.

**Weaknesses:**

**Abstract Treatment of Memory Mechanisms**
While Definition 4.7 defines a memory mechanism as a mapping from base context K to effective context K_eff, it does not differentiate between implementation strategies (e.g., external memory, world models, state-space models). A deeper discussion of how different architectures realize µ(K) would strengthen the framework.

**Limited Coverage of Other Memory Types**
The paper focuses primarily on declarative memory along the temporal axis. Although the authors explicitly state they do not aim to replicate all aspects of human memory, a brief discussion of how “working memory” or “episodic memory” might fit into this taxonomy would enhance its extensibility.

The paper reads more like a taxonomy of memory types and seems to lack sufficient novelty or creativity.

**Questions:**

NO

---

> ### Author Response · Authors · 2025-11-22
> **Ofifcial comment by Authors**
>
> We thank Reviewer vvf9 for their efforts in reviewing our work. In particular, we are grateful for pointing out that the proposed formalization fills a significant void in the current RL literature, where “memory” is often used loosely or inconsistently.
>
> > W1. Abstract Treatment of Memory Mechanisms
>
> Thanks for your comment. Our definition of the memory mechanism $\mu(K)$ in Definition 4.7 intentionally abstracts away implementation details, since the framework applies to any architecture whose effective context length satisfies $K_{eff}=\mu(K)\geq K$. However, we agree that clarifying how different architectures instantiate $\mu(K)$ would strengthen the exposition.
> In the original version of the article, Appendix C - Memory Mechanisms discusses the main methods for implementing memory in RL agents: recurrent neural networks, transformers, state-space models, temporal convolutions, world models, graph neural networks, external memory buffers, and autostigmergy. These approaches are fundamentally different in nature, and therefore including all implementation-level distinctions directly inside Definition 4.7 would break the architectural agnosticism that the framework is designed to preserve. Instead, Definition 4.7 formalizes a necessary and sufficient functional criterion for long-term memory, irrespective of whether $\mu(K)$ is realized via recurrent state accumulation, external read/write operators, latent belief-state propagation, or explicit retrieval mechanisms.
>
> > W2. Limited Coverage of Other Memory Types. Although the authors explicitly state they do not aim to replicate all aspects of human memory, a brief discussion of how “working memory” or “episodic memory” might fit into this taxonomy would enhance its extensibility.
>
> We thank the Reviewer for this insightful remark.
> We already had a brief discussion of how "working memory" or "episodic memory" works in the original version of this paper (see Section 2.1, L079-082).
> We consider a more in-depth study of the concepts of "working memory" or "episodic memory" as future work, which we explicitly address in Section 6, Conclusion (L519-520).
>
> > The paper ... lack sufficient novelty or creativity.
>
> While we understand the concern that the work may appear primarily taxonomic, we would like to clarify that the contribution is not limited to classification. For example, we highlight the problem of the lack of unified memory taxonomy in RL and demonstrate experimentally that this can lead to subtle poor agent evaluation results. In detail, the following main features of our work can be highlighted:
>
> First, as the Reviewer notes in the Strengths section, the paper “successfully translates ambiguous cognitive science terms into precise, quantifiable, and verifiable definitions… filling a significant void in current RL literature.” We view this as evidence that the contribution goes beyond a descriptive taxonomy: the framework formalizes previously informal terminology into measurable quantities grounded in POMDP theory (Definitions 4.4-4.6).
>
> Second, although the concepts of STM, LTM, declarative, and procedural memory appear in prior work, to the best of our knowledge no existing work provides formal mathematical conditions for when an RL agent requires long-term memory, nor a provable separation between STM/LTM regimes based on quantitative parameters. The theoretical results (Theorem 2), together with the notion of memory-intensive environments and the context memory border, enables falsifiable evaluation of memory,  absent from prior literature, where memory competence is often inferred from architectural intuition rather than measurable properties.
>
> Third, the framework is operational rather than descriptive: Algorithm 1 prescribes how to construct STM-only, LTM-only, or mixed memory regimes inside a single POMDP, and Section 5 demonstrates that widely used “memory-enhanced” agents fail under correct LTM configurations even when they appear to succeed under standard evaluation. These results are not obtainable from a taxonomy alone.
>
> Finally, we agree that the paper introduces a taxonomy, but it exists because a rigorous evaluation methodology requires one. The taxonomy is therefore not the final product of the work, but a prerequisite for the theoretical and empirical results that follow.
>
> We sincerely appreciate the Reviewer’s thoughtful feedback and the careful attention devoted to our submission. Their comments have substantially improved the clarity and quality of the manuscript, and we hope the revisions make the contributions and technical results fully transparent. We believe the strengthened version addresses all concerns comprehensively, and we remain available for any further discussion or clarification that might assist in the final evaluation.

---

### Official Review · Reviewer_jjn4 · 2025-11-01

**Soundness:** 3
**Presentation:** 2
**Contribution:** 3
**Rating:** 4
**Confidence:** 3

**Summary:**

This paper highlights the lack of a unified methodology for validating “memory’ for RL agents and presents a robust experimental methodology and standardize evaluations. In particular, this paper provides formal definitions to clarify the widely adopted concepts of memory in cognitive science.

**Strengths:**

This paper is generally well-written and well-motivated. This paper studies the relatively underexplored subject of “memory in RL agents” and presents formal definitions of various memory concepts, highlighting the importance of appropriate experimental configurations for evaluating them. Pictorial illustrations help clarify the concepts.

**Weaknesses:**

Some of the definitions require more explanation to fully understand the concept. Paper formatting could be improved. Please see the questions and comments.

**Questions:**

**Questions**

Q1. Is it possible to present the outer loop in Figure 3?

Q2. Could the authors provide a more detailed explanation of $n$ in $\{\zeta_n\}_n$? Do we also have a set of $n$ here? This point is not very clear. $n$ should be properly defined first.

Q3. Please use consistent notation for $\min_n \Theta$ and $\max_n \Theta$, instead of mixing $\min_n \Theta$ and $\min \Theta$.

Q4. Some illustrations depicting cases that validate LTM and STM for different $K$ values (as explained in the lower part of Page 6) would help clarify the general characteristics.

**Comments**

C1. Please check the citation format in “skills across tasks Team et al. (2023)” on Page 2.

C2. Figure 4 could be properly defined as a table rather than a figure. The term 0-shot seems a bit unnatural. Please use a more common term, such as 'zero-shot'.

C3. Please check the citation format for Theorem 2 on Page 5 and for Algorithm 1 on Page 7.

C4. $L$ is used before being properly defined, and its definition is placed within the script for Table 1. The reviewer strongly recommends defining the values before use.

---

> ### Author Response · Authors · 2025-11-22
> **Ofifcial comment by Authors**
>
> We thank Reviewer jjn4 for their comments on our work. In particular, we are grateful that the Reviewer noted that our work studies the relatively underexplored topic of "memory in RL agents".
>
> > Q1. Is it possible to present the outer loop in Figure 3?
>
> In our formulation, the term Meta-RL denotes the class of problems in which learning occurs across tasks through an explicit outer-loop optimization over experience distributions. This structural separation allows us to disentangle Meta-RL from the Memory-DM setting, where all learning and memory usage occur within a single environment and episode.
>
> Our framework is general enough to characterize both declarative and procedural memory. In particular, procedural memory arises in Meta-RL whenever the inner-loop must acquire and reuse skills across episodes or environments. Inside the inner-loop, one may further distinguish short-term and long-term forms of memory by applying the same correlation-horizon criterion used in Memory-DM. However, extending this STM/LTM distinction to the outer-loop would require introducing a second temporal scale associated with adaptation across lifetimes, which is not addressed in the present work.
>
> For this reason, Figure 3 intentionally excludes the outer-loop. Its purpose is to classify within-episode memory dependencies in Memory-DM tasks. The broader taxonomy that connects Memory-DM and Meta-RL, including the role of the outer-loop, is instead summarized in Table 1 (formerly Figure 4), where we explicitly mark which regimes require procedural memory and which do not.
>
> This separation is deliberate: the proposed framework targets declarative memory in Memory-DM as the primary object of study, while acknowledging that Meta-RL entails procedural memory but leaving its long- vs. short-term structure for future work.
>
> > Q2. Could the authors provide a more detailed explanation of $n$ in $\zeta_n$? Do we also have a set of $n$ here? This point is not very clear. should be properly defined first.
>
> In the text of our paper, $\zeta$ **does not appear**. You must have meant $\xi_n$, that is, the correlation horizon (the temporal distance in the event-recall pair) associated with the $n$-th such pair. All horizons collectively form the set $\Xi = \{\xi_n\}_{n=1}^N$, where $N$ is the total number of event-recall dependencies in the environment, $t_e^n$ is the event start time, $\Delta t^n$ its duration, and $t_r^n$ the time at which the agent must recall that event when making a decision. Thus, yes, we have a set of $\xi_n$, one per event-recall dependency. We have clarified this in the text to make the idea clearer.
>
> > Q3. Please use consistent notation for $min_n \Theta$ and $max_n \Theta$, instead of mixing $min_n \Theta$ and$ \min \Theta$.
>
> In the text of our paper, **neither** $min_n \Theta$ and $max_n \Theta$, nor $min_n \Theta$ and$ \min \Theta$ **occur.** We assume you meant $min_n \Xi$ and $max_n \Xi$ instead. By default, since $\Xi = \{\xi_{n}\}_{n=1}^{N}$, it is assumed that the $min$ and $max$ operations are used relative to the index of the corresponding event-recall pair (correlation horizon), but for notational simplicity, we omit this index $n$ in some cases, although we continue to keep it in mind. We have corrected this in the updated version of the paper, and now explicitly write out the index $n$ everywhere.
>
> > Q4. Some illustrations depicting cases that validate LTM and STM for different $K$ values (as explained in the lower part of Page 6) would help clarify the general characteristics.
>
> Our text already contains a corresponding illustration (Figure 2) that demonstrates how LTM and STM are defined in our work with respect to $\xi$ and $K$.
>
> > C1. Please check the citation format in “skills across tasks Team et al. (2023)” on Page 2.
>
> Thank you for your comment, we have corrected this in the updated version of the text.
>
> > C2. Figure 4 could be properly defined as a table rather than a figure. The term 0-shot seems a bit unnatural. Please use a more common term, such as 'zero-shot'.
>
> Thank you for your comment, we have corrected this in the updated version of the text.
>
> > C3. Please check the citation format for Theorem 2 on Page 5 and for Algorithm 1 on Page 7.
>
> Thank you for your comment, we have corrected this in the updated version of the text.
>
> > C4. $L$ is used before being properly defined, and its definition is placed within the script for Table 1. The reviewer strongly recommends defining the values before use.
>
> $L$ refers to the corridor length in the Passive T-Maze task. We've added its definition to the updated text.
>
> We are once again grateful to Reviewer jjn4 for their questions and comments on our work. We have responded to all questions, corrected all typos and ambiguities in the text, and are open to further discussion.

---

### Author Response · Authors · 2025-12-02
**General Response by Authors**

We sincerely appreciate the Reviewers’ thoughtful feedback and their recognition of our work as a meaningful step toward clarifying and formalizing the role of memory in Reinforcement Learning. Their comments affirm the contributions of our paper, and we are pleased that they find the motivation, definitions, taxonomy, and analyses valuable. Below, we summarize the key strengths highlighted in the reviews:
1. **Well-written and well-motivated paper** with clear organization and accessible explanations of core ideas (Reviewers jjn4, Wrt1)
2. **Clarification of the underexplored topic of memory in RL** through precise problem framing and a systematic examination of how memory should be conceptualized and evaluated in RL agents (Reviewer jjn4).
3. **Effective visual illustrations** that help clarify abstract memory concepts and their operationalization (Reviewer jjn4).
4. **Rigorous translation of cognitive-science terminology** into precise, measurable, and verifiable constructs for RL, addressing longstanding ambiguity in how terms such as short-term/long-term and declarative/procedural memory are used (Reviewer vvf9).
5. **Unified evaluation framework for memory types** based on the distinction between Memory Decision-Making and Meta-RL, and on operational criteria for declarative and procedural memory using $(n_{\text{envs}}, n_{\text{eps}})$ configurations (Reviewer vvf9).
6. **Standardized and fair definitions of memory categories** providing a consistent basis for comparing future memory-augmented RL models (Reviewer RqbQ).
7. **Careful and rigorous formalization** of memory types, context length, and context–memory boundaries (Reviewer RqbQ).
8. **Clear taxonomy of memory in RL** including well-defined distinctions between short-term and long-term memory and mathematically grounded notions such as the context memory border (Reviewer Wrt1).
9. **Insightful experimental demonstration** showing that variable correlation horizons in common RL benchmarks can mask long-term memory deficits, emphasizing the need for principled memory evaluation (Reviewer Wrt1).

Across the revised version and our responses, we addressed the main concerns by:
1. clarifying and tightening all definitions, notation, and figure / table layout;
2. expanding the discussion of concrete memory mechanisms and their instantiations within our abstract framework;
3. sharpening the positioning of the paper as a foundational, operational framework for defining and testing memory in RL, supported by targeted experiments that concretely demonstrate how current evaluation practices can lead to misleading conclusions.

We are grateful once more for the Reviewers’ thoughtful comments. We believe our answers and the revisions directly resolve the outstanding concerns and lead to a significantly improved paper.

---

### Meta-Review · Area_Chair_pTMm · 2026-01-07

**Summary:**

1. Requests for more clarity / explanations.
2. Paper is just a taxonomy and thus not novel enough.
3. Applicability of insights to practical problems / RL deployments.

**Reviewer Concerns:**

1. Clarity concerns were addressed.
2. Good taxonomies can be useful.
3. Applicability of the insights is somewhat limited, but the proposed framework / taxonomy can be helpful for people in the ICLR community to at least get an overview of the area.

**Reviewer Scores:**

1. jjn4: 4->6 (clarifications provided)
2. vvf9: 4->4 (reviewer unlikely to be convinced on novelty)
3. RqbQ: 4-> 6 (concerns about LSTM vs transformers addressed well).
4. Wrt1 : 6->6 (concerns partially addressed, but  there still remains a difference in opinion on novelty between the reviewers and the authors)

---

### Decision · Program_Chairs · 2026-01-26

Accept (Poster)